# Indirect Nuclear Magnetic Resonance (NMR) Spectroscopic Determination of Acrylamide in Coffee Using Partial Least Squares (PLS) Regression

**Vera Rief [1,2], Christina Felske [1], Andreas Scharinger [2], Katrin Krumbügel [2,3], Simone Stegmüller [1], Carmen M. Breitling-Utzmann [4], Elke Richling [1], Stephan G. Walch [2] and Dirk W. Lachenmeier [2,*]**

1   Department of Chemistry, Food Chemistry and Toxicology, Technische Universität Kaiserslautern, Erwin-Schrödinger-Straße 52, 67663 Kaiserslautern, Germany; vera.rief@mailbox.org (V.R.); christina_felske@web.de (C.F.); stegmueller@chemie.uni-kl.de (S.S.); richling@chemie.uni-kl.de (E.R.)
2   Chemisches und Veterinäruntersuchungsamt (CVUA) Karlsruhe, Weissenburger Strasse 3, 76187 Karlsruhe, Germany; andreas.scharinger@cvuaka.bwl.de (A.S.); katrin.krumbuegel@hotmail.de (K.K.); stephan.walch@cvuaka.bwl.de (S.G.W.)
3   Rheinische Friedrich-Wilhelms-Universität Bonn, Regina-Pacis-Weg 3, 53113 Bonn, Germany
4   Chemisches und Veterinäruntersuchungsamt Stuttgart, Schaflandstr. 3/2, 70736 Fellbach, Germany; carmen.breitling-utzmann@cvuas.bwl.de
*   Correspondence: lachenmeier@web.de; Tel.: +49-721-926-5434

**Abstract:** Acrylamide is probably carcinogenic to humans (International Agency for Research on Cancer, group 2A) with major occurrence in heated, mainly carbohydrate-rich foods. For roasted coffee, a European Union benchmark level of 400 µg/kg acrylamide is of importance. Regularly, the acrylamide contents are controlled using liquid chromatography combined with tandem mass spectrometry (LC–MS/MS). This reference method is reliable and precise but laborious because of the necessary sample clean-up procedure and instrument requirements. This research investigates the possibility of predicting the acrylamide content from proton nuclear magnetic resonance (NMR) spectra that are already recorded for other purposes of coffee control. In the NMR spectrum acrylamide is not directly quantifiable, so that the aim was to establish a correlation between the reference value and the corresponding NMR spectrum by means of a partial least squares (PLS) regression. Therefore, 40 commercially available coffee samples with already available LC–MS/MS data and NMR spectra were used as calibration data. To test the accuracy and robustness of the model and its limitations, 50 coffee samples with extreme roasting degrees and blends were additionally prepared as the test set. The PLS model shows an applicability for the varieties *Coffea arabica* and *C. canephora*, which were medium to very dark roasted using drum or infrared roasters. The root mean square error of prediction (RMSEP) is 79 µg/kg acrylamide (*n* = 32). The current PLS model is judged as suitable to predict the acrylamide values of commercially available coffee samples.

**Keywords:** 2-propenamide; *Coffea arabica*; multivariate data analysis; spectroscopy; LC–MS/MS; coffee roasting

## 1. Introduction

Research on the formation and biological effects of acrylamide is a relatively young field. In 1994 acrylamide was classified as probably carcinogenic to humans (group 2A) by the International Agency for Research on Cancer (IARC) [1]. In a study by Bergmark et al. in 1997, unexpectedly high exposures were found in non-smokers that could not be explained. Until then, high exposure was considered likely only for workers in polyacrylamide gel production and heavy smokers [2]. In 2000, the research group led by Tareke et al. was able to observe a correlation between fried food and increased acrylamide exposure [3] and subsequently showed in 2002 that specifically heated carbohydrate-rich foods have high acrylamide contents [4]. In 2004, the assumption that acrylamide is formed by the Maillard

reaction pathway was confirmed [5,6]. Cereal, potato, and coffee products are subject to European Union (EU) Regulation 2017/2158, which sets benchmark levels for various food groups in addition to minimization the formation of acrylamide [7]. In 2019, roasted coffee contained an average of 195 µg/kg acrylamide [8], and the benchmark level is currently 400 µg/kg acrylamide [7]. For the determination of acrylamide content, the coupling of liquid chromatography with tandem mass spectrometry (LC–MS/MS) is considered as the standard method [9]. This approach is time-consuming due to the necessary sample preparation and measurement procedure [4]. A possible alternative is the recording of nuclear magnetic resonance spectra (NMR). However, acrylamide cannot be quantified directly from NMR spectra because its concentration is typically below the detection limit of the acrylamide-specific resonances [8]. The aim of this research was to use a partial least squares (PLS) regression model that allows a possible correlation between reference results from the LC–MS/MS method and associated NMR spectra. The established PLS model could then be used to calculate acrylamide content indirectly from NMR spectra. A data set of 40 commercially available coffee samples was available for the establishment of the PLS model. In order to test the predictive accuracy and limitations of the model, 50 coffee samples are additionally roasted, some of which correspond to commercially available coffees and some of which have extreme roasts and blends of varieties.

## 2. Results

### 2.1. Calibration of the PLS Model

#### 2.1.1. Data Preprocessing

The y-data must be at least mean centered to be suitable for use in the PLS models [10]. Further standardization of the y-data does not change the result. The x-data can be used in the PLS toolbox unprocessed, mean centered or standardized. If the x-data are used unprocessed, the regression error was highest (root mean square error of cross validation, RMSECV = 106 µg/kg) and the sum of the explained variance was lowest (59.1%). This can be slightly improved by mean centering (RMSECV = 105 µg/kg, 62.1%). Standardizing the x-data reduced the RMSECV drastically (RMSECV = 71 µg/kg) and 99.9% of the total variance was achieved with six PCs. In addition, a weighting of the buckets was produced, revealing the influence of the posterior spectrum region. In Figure 1, left side the loadings of each bucket were plotted for each PC. The higher the magnitude of this value, the higher the influence of the variable on the respective PC. For mean centered x-data, buckets in the first half show an especially high influence on PC1. Buckets between number 80 and 90 had a particularly negative loadings value at −5.5. In the second half, the loadings for all PLS components were very low, so they had little influence on the regression result. The loadings for the other PCA components PC2 to PC6 were generally very low. If the x-data were standardized, the influence of the posterior increased visibly (Figure 1b). There was still a negative correlation with PC1, but the loadings became larger in terms of magnitude. Buckets after number 950 show a positive correlation with PC1. In general, influences on the other PCs can be seen. In the following, mean centered y-data and standardized x-data were used.

#### 2.1.2. Variable Selection

In the PLS regression, it is assumed that the x-data contain discriminating variables to explain the y-data. Variables that had little information, on the other hand, increased the background noise and leads to a larger regression error. In addition, the result was easier to understand and interpret with fewer variables.

Five different methods for variable selection are available in the PLS toolbox. Each variable selection aimed to minimize the RMSECV by excluding variables without information (background noise) in the PLS calibration. Table 1 shows the selection results of the different methods. In each case, the specifications of a method were chosen so that the RMSECV reached a minimum. A minimum of 19 and a maximum of 315 buckets were selected from a total of 1042 buckets. In general, a reduction of the RMSECV could

be observed by using variable selection, which was 71 µg/kg without variable selection. Bucket no. 877 and no. 904 were selected by all methods, so they were likely to have a high impact on the regression result. Most of the selected buckets were in the posterior spectrum region (after no. 800).

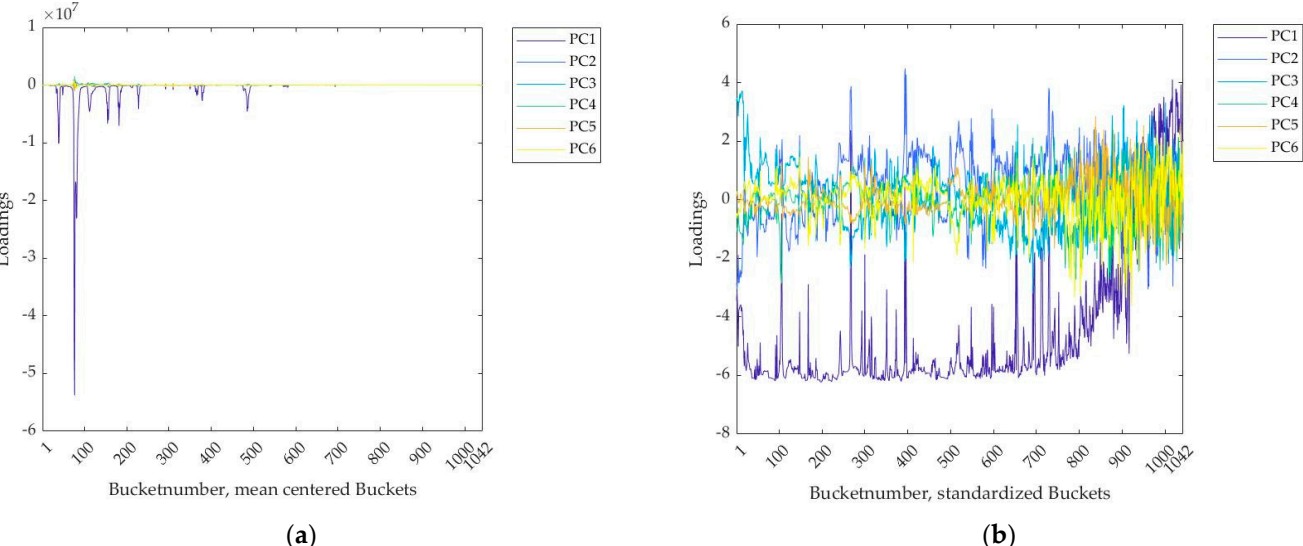

**Figure 1.** (**a**) Loadings from each bucket for every PLS component (PC), PLS regression (6 PLS components) with 40 commercial coffee samples and 1042 mean centered buckets for each sample; (**b**) loadings for each PLS component, PLS regression (6 PLS components) with 40 commercial coffee samples and 1042 standardized buckets each.

**Table 1.** Comparison of variable selections based on PLS regression of mean-centered y-data and standardized x-data with six PLS components, PLS toolbox.

| Method | Specification | Number of Selected Buckets | RMSECV (µg/kg) |
|---|---|---|---|
| GA | Interval size 5 | 170 | 38 |
| Forward iPLS | Interval size 1 | 23 | 18 |
| rPLS | Specified, Level 6 | 36 | 28 |
| | suggested | 19 | 35 |
| | surveyed | 32 | 19 |
| sRatio | Exclusion of last 45% | 315 | 48 |
| VIP | Exclusion of last 20% | 140 | 36 |

The genetic algorithm (GA) and forward interval PLS (iPLS) selected different buckets depending on the selected interval size. Table 1 shows the interval sizes with the smallest error. In addition, the GA did not give reproducible results due to the evolutionary biology background (combinations and mutations of variables in the subset), making it unsuitable for this work. Forward iPLS gave a very small RMSECV, but the model was based on only 23 buckets, so these would be risky as the basis of a prediction model. If the signals of a more unusual coffee (e.g., extreme degree of roast, special variety) do not match with the 23 selected buckets, an incorrect prediction is made. For the reverse PLS (rPLS), the "specified" variant yields the smallest RMSECV with the largest number of variables, so that the "suggested" and "surveyed" variants were not considered for this work. Selectivity ratio (sRatio) was also considered unsuitable due to the high error and the high number of variables. Selected buckets from the variable of importance (VIP) method and specified rPLS were then compared and chemical compounds were identified. Especially in the rear region (from about 10 ppm), hardly any signals could be seen, which made the identification of selected buckets in this region difficult. Specified rPLS selected fewer buckets than VIP, which can be seen well in the front region of the spectrum (<7 ppm). Here, VIP selected many signals that can be assigned to lipids (sterols, diterpenes, fatty acids, and glycerol) and were thus uninteresting for Maillard formation of acrylamide.

Exclusion of these signals (36 buckets in total) improved the RMSECV from 36 to 35 µg/kg. Specified rPLS selected only the signals from 16-*O*-Methylcafestol (OMC) (δ: 3.164 ppm (s) and 4.423 ppm (d, *J* = 12.55 Hz) [11,12]). In the posterior region of the spectrum, furfuryl alcohol (δ: 7.401 ppm (dd, *J* = 0.82, 1.88 Hz) [13,14]), hydroxymethylfurfural (HMF) (δ: 9.616 ppm (s) [13,14]), and an Arabica-specific signal (δ: 10.182 ppm (d, *J* = 7.3 Hz) [11,15]) can be identified.

### 2.1.3. Spiking

To further identify selected buckets, spiking experiments were performed with the roasting markers, namely furan, 2-furoic acid, methyl nicotinate and the acrylamide precursor *L*-asparagine. It is observed that *L*-asparagine was insoluble in deuterated chloroform (CDCl$_3$) and thus, no NMR spectrum could be recorded. Furan can be identified from the triplets at 6.397 ppm and 7.452 ppm. 2-furoic acid shows signals at 6.548 ppm (m), 7.280 ppm (dd) and 7.625 ppm (q). In addition, a singlet at 8.103 ppm can be assigned to both furan and 2-furoic acid. Methyl nicotinate had signals at 3.964 ppm (s), 7.398 ppm (ddd), 8.304 ppm (dt), 8.781 ppm (dd), and 9.233 ppm (dd). There was no match with the signals from the coffee samples besides the furan (derivative) signal. By adding the singlet at 8.103 ppm (bucket no. 752), the RMSECV could be lowered to 26 µg/kg.

### 2.2. Parameters of the PLS Model

In the considerations above, the change in RMSECV was the main parameter. By centering the y-data and standardizing the x-data, the lowest RMSECV of 71 µg/kg could be achieved. Further reduction of the RMSECV was achieved by targeted variable selection with the specified rPLS method, so that the RMSECV was 28 µg/kg. With spiking the signal at 8.103 ppm could be assigned to furan (derivative). Adding the associated bucket reduces the RMSECV to 26 µg/kg. The coefficient of determination was $R^2$ = 0.94. The six PLS components together explained 99.3% of the variance in the data. A total of 77.5% explained variance fell to the first PLS component. The PC2 explained 18.22% and PC3 2.23% of the variance. The last three PLS components hardly contributed to the explanation of the variance (less than 1% each).

The influence of the variables can be shown in the scores and loadings plots for the first two PLS components. In the scores plot, data points in the upper right square had above average influence in PC1 and PC2, whereas data points in the lower left square had below average influence in both PLS components [10]. The 40 samples with their attributes (variety, roasting degree, organic coffee, and cardamom addition in Turkish coffee) are shown on the left side in Figure 2. The assignment to the variety *Coffea canephora* was based on high OMC contents (>1000 mg/kg) of the sample. For some samples, no other attributes were known besides varieties, so there was only limited significance. Dark roasted *C. arabica* were mainly located in the upper left square, meaning they were below average in PC1 and above average in PC2. Samples of *C. canephora* were located on the right side of the scores plot regardless of roasting degree and were thus above average with respect to PC1. Only two samples had the attribute "light roast" assigned and both were in the upper right square regardless of their variety. The organic coffees (all *C. arabica*) revealed a group that was slightly below average in PC1 and PC2. The addition of cardamom in Turkish coffee led to a high above average influence with respect to PC2. In Figure 2b, the samples were marked with their acrylamide content (µg/kg). The 40 samples contained 23 different concentrations ranging from 95 to 490 µg/kg acrylamide. The six samples with the highest acrylamide content were in the upper right square, so they were above average with respect to both PLS components. The seven samples with lowest content were located in the lower left square, so they were below-average for both PLS components. A separation into samples with higher (250–480 µg/kg) and lower (95–210 µg/kg) acrylamide content might even be visually possible.

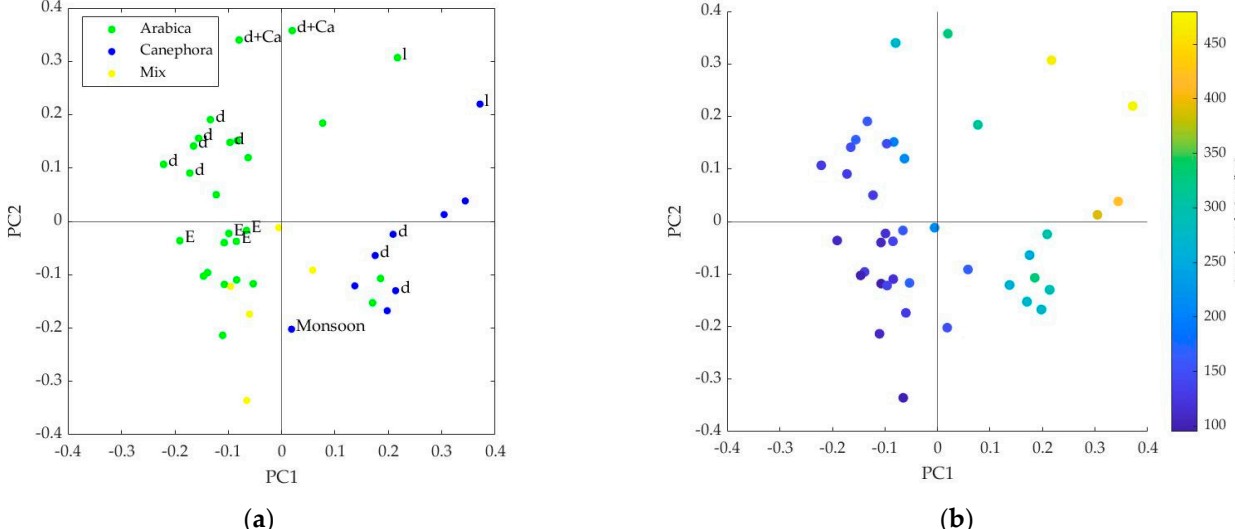

(**a**)                                                    (**b**)

**Figure 2.** (**a**) Scores plot of principal components (PC) 1 and 2, 40 commercial coffee samples are marked with their known attributes: l=light roasting degree, d= dark roasting degree, +Ca= additive cardamom in Turkish coffee; (**b**) scores plot of PC1 and PC2, 40 commercial coffee samples are marked with their measured content of acrylamide [µg/kg].

In the loadings plot, data points in the upper right square had a positive high impact on both PLS components, while in the lower left square they had a negative high impact [10]. In Figure 3, it can be seen that only bucket no. 900 (unknown), no. 966 (unknown), and no. 1018 (unknown) were located in the lower left square. In the upper right square were the buckets no. 393 (OMC) and no. 394 (OMC) and seven other buckets with an unknown compound. The buckets at the back of the spectrum (>no. 1000) all had a negative impact on PC2 and, with the exception of bucket 1018, a positive impact on PC1. The additional bucket 752 from the furan signal (8.103 ppm, s) was located just inside the upper left square and thus had a negative influence on PC1 and a small positive influence on PC2.

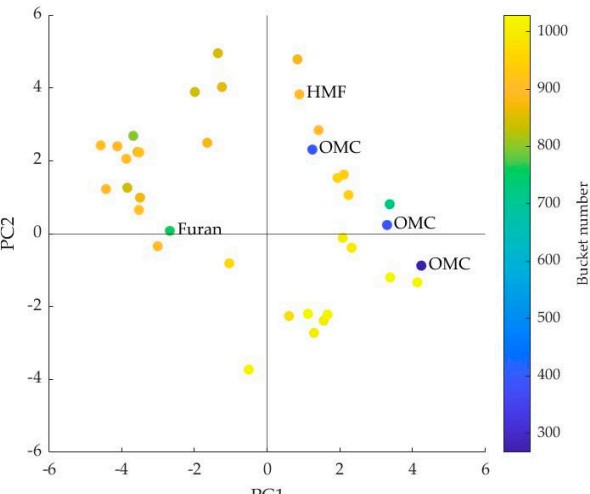

**Figure 3.** Loadings plot of principal component (PC) 1 and 2; identified buckets are marked with chemical compound.

### 2.3. Application of the PLS Model to Unknown Data

The model was based on 40 commercially available coffee samples. Most of these are of the variety *C. arabica*, and usually provided a dark roasting degree. In order to test the model and show its limitations, more samples with extreme roasting were prepared and afterwards predicted. For this purpose, *C. canephora* beans and Malabar and Catuaí types of *C. arabica* were roasted in an infrared (IR) roaster from very light to very dark. In all roasts,

an increase in volume of the roasted beans can be observed and the characteristic cracking of the beans can be heard. In addition, more samples of *C. arabica*, Catuaí were used, which were roasted in a drum roaster. In total, a test set of 50 samples was assembled, their NMR spectra recorded, and their acrylamide contents determined via the standard LC–MS/MS method. The results from the model prediction and the reference measurements can be taken from Appendix A (Tables A1 and A2). It must be noted that the acrylamide contents from the reference samples were subject to fluctuations and were used here as the mean value of a double determination. The acrylamide contents provided by the standard LC–MS/MS method were between 16 and 1300 µg/kg, those from the resulted model prediction between 31 and 380 µg/kg.

Including all test set samples, the correlation was $R^2 = 0.33$ and the RMSEP (root mean square error of prediction) was 334 µg/kg. For comparison, within the calibration set (cross-validation) an $R^2 = 0.94$ and an RMSECV = 26 µg/kg applies. It should be noted, that using the $R^2$ for assessing the prediction accuracy for unknown data was not recommended. The focus was on the behavior of the RMSEP [16]. Accurate predictions were made for a very dark *C. canephora* (1 µg/kg above the reference value), a dark *C. arabica*, Malabar and a dark *C. canephora* (each 8 µg/kg above the reference value).

In Table 2, the samples were sorted according to the roasting degree, variety, and roasting method in order to identify a possible influence of these parameters. The largest RMSEP are found in the light roasts regardless of variety and roasting method. One particular sample is *C. arabica*, Catuaí, which was roasted (IR) for only 6.5 min and is therefore more like a green coffee than a roasted coffee. It contained 16 µg/kg acrylamide according to the reference LC–MS/MS measurement and was predicted to contain 308 µg/kg. A light roasted sample (*C. arabica*, Catuaí) had the highest acrylamide content with 1300 µg/kg measured and 380 µg/kg predicted. When sorted by variety, the most accurate predictions were achieved for *C. canephora* samples. It can be seen by LC–MS/MS measurements that for mixtures with increasing *C. canephora* content, the acrylamide content increased. Such a trend was not observed by the model prediction for light roasts, but for dark roasts. In Figure 4, the samples are color-coded according to their degree of roasting. For one sample, the degree of roasting was not known (circle). If all light roasts were excluded, a correlation of $R^2 = 0.79$ was obtained and the RMSEP decreased to 79 µg/kg, which were judged as acceptable for application in the form of a screening analysis to preselect suspicious samples.

**Table 2.** RMSEP (µg/kg), grouped by roasting degree, type and roaster.

| | RMSEP (µg/kg) |
|---|---|
| **Roasting degree** | |
| Light (*n* = 18) | 546 |
| Medium (*n* = 10) | 52 |
| Medium/dark (*n* = 2) | 87 |
| Dark (*n* = 14) | 98 |
| Very dark (*n* = 5) | 53 |
| **Type** | |
| *C. arabica*, Catuaí (*n* = 19) | 97 |
| *C. arabica*, Malabar (*n* = 4) | 48 |
| *C. canephora* (*n* = 5) | 17 |
| **Roaster** | |
| IR (*n* = 26) | 49 |
| Drum (*n* = 6) | 151 |

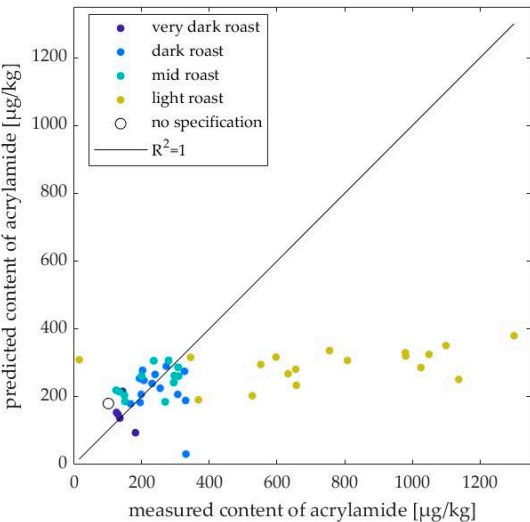

**Figure 4.** Correlation between the predicted content of acrylamide (μg/kg) and the measured content of acrylamide (μg/kg) using LC–MS/MS; coffee samples are color-coded with their roasting degree.

## 3. Discussion

### 3.1. Interpretation of the PLS Components

Of greatest importance are the first two PLS components, as they explain most of the variance in the data. In the scores plot, PC1 and PC2 can be used to separate samples with higher (250–480 μg/kg) and lower (95–210 μg/kg) acrylamide levels.

PC1 was above average especially in *C. canephora* samples, since *C. canephora* contains on average more acrylamide than *C. arabica*. In addition, the two light roasts are above average in PC1, as the acrylamide content decreases with increasing degree of roasting [8]. The extrema on PC1 are a dark *C. arabica* roast at the minimum and a light *C. canephora* roast at the maximum. In the loadings plot, besides some unknown compounds, OMC and HMF had a high positive influence. For HMF, there are sources describing a negative correlation with acrylamide [8], and sources describing a positive correlation [17,18]. In this model, HMF is located in the upper right square and thus is positively correlated with acrylamide (PC1). In addition, it has a positive influence on PC2. HMF was selected as important in all available variable selection methods. Therefore, it has an important role in predicting acrylamide content. Furan correlates negatively with acrylamide content. The darker the roast, the higher the furan content [8]. In the loadings plot, this is evident at the position just inside the upper left square, which represents a negative influence on PC1 and a slightly positive influence on PC2. PC1 explained 77.5% of the variance in the data and was likely based on variety and roasting degree. The lighter the roast (higher HMF content, less furan) and the higher the *C. canephora* content (increasing OMC content), the more acrylamide was present in the sample. The importance of PC1 is consistent with the literature [8,17,18].

With the data available, the influences of PC2 cannot be clearly substantiated. Many samples have no known attributes other than the variety (important for scores plot) and only 5 out of 37 buckets could be assigned to chemical compounds (important for loadings plot). The highest loadings for PC2 were achieved by dark *C. arabica* roasts with additional cardamom, followed by the two light roasts. The lowest loading had a mixed sample with 60% *C. arabica* content. No clear separation in terms of roasting degree or variety was evident. For the explanation of PC2, the identification of the compound behind Bucket 877 would be very interesting. This bucket had the highest positive influence on PC2 and was selected by all available selection methods. Thus, it seemed to be a key compound for acrylamide formation.

### 3.2. Applicability of the PLS Model

The model is calibrated using 40 commercially available coffee samples. They show a medium to darker roasting degree. Special roasts, such as cinnamon/Scandinavian (light) or Neapolitan (dark), are often highlighted on the packaging. Roasting is mostly performed in hot air and drum roasters. In addition, they are predominantly pure *C. arabica* or blends thereof [19]. Therefore, the application to deviating coffee samples was problematic because the calibration data did not cover the range of data for prediction.

Based on Figure 4 and the RMSEP values, the strong deviation of the prediction for light roasts was obvious. Only two of 40 samples were known to be light roasts for the purpose of calibrating the model, so there was not a sufficient basis for predicting light roasts. Since the model was created with samples below the guideline value of 400 µg/kg, it was also not suitable for samples with higher acrylamide contents, as in light roasts. Excluding the light roasts significantly improved the overall prediction.

When sorted by variety, the low RMSEP of 17 µg/kg for *C. canephora* samples was noticeable. In the model, samples with an OMC content above 1000 mg/kg were assigned the attribute "*C. canephora*"; a more precise declaration was not available. It was shown that variety separation occurs on PC1, which probably led to the very good prediction of the *C. canephora* samples.

Despite the roasting method differing from the model, the IR roasts can be predicted well. For the drum roasts, the RMSEP was higher because the model predicted an acrylamide content of 31 µg/kg for a dark *C. arabica*, Catuaí, although the reference value was 331 µg/kg. The other two dark *C. arabica*, Catuaí drum roasts had more precise predictions of 189 µg/kg and 207 µg/kg acrylamide. It is unclear why this one sample was assigned an acrylamide content comparable to almost unroasted green coffee. In general, the model made the most accurate predictions for medium to very dark *C. canephora* samples roasted in the IR roaster.

Apart from the low variance of commercial coffees, which were typically medium roasted *C. arabica*, limitations of the research included the restricted number of samples in the training set, limited by availability and costs of reference analytics. Nevertheless, as a validation with a completely independent test set was conducted, limited applicability of the current model for commercial roasts can be assumed, while rather large errors were observed for light and dark roasts. To overcome this problem, an expanded training set, combined with identification of important NMR signals influencing PC2, could be conducted.

### 3.3. Similar Applications in Literature

In literature, principal component analysis (PCA) has often been used for classification purposes. Monakhova et al. were able to determine varietal authenticity using NMR spectra of aqueous and lipophilic green coffee extracts. In aqueous NMR spectra, differences in caffeic and quinic acid contents were seen, whereas in lipophilic spectra the compounds OMC and kahweol were discriminating. Clear separation of the varieties in the loadings plot was seen for both types of preparation, so the application of PCA to NMR spectra was suggested as a rapid screening method for authenticity testing [11]. PCA could also be used for geographical assignment of roasted coffee. For this purpose, samples of *C. arabica* with known origin (America, Africa, and Asia) were measured as aqueous extracts in NMR and a separation by place of origin in the scores plot was achieved. Classification was achieved based on chlorogenic acid and lactate (Africa), acetate and trigonelline (Asia), and fatty acid protons (America) [20]. In addition to assigning the variety and country of origin, Wei et al. used the PCA method to classify the roasting time and thus the degree of roasting. Discriminatory compounds included chlorogenic and quinic acid, sucrose and HMF, and trigonelline and nicotinic acid. There was a correlation between the respective origin compound in green coffee and the thermal degradation product with increasing degree of roasting [18].

Febvay et al. used PLS regression to determine roast markers in NMR spectra of aqueous coffee extracts. For this purpose, samples with different final roasting temperatures and the same roasting time and samples with the same final temperature and different roasting times were roasted. To test the influence of the final temperature, a model with 13 components (explained variance 97.8%) was built and cross-validated with the leave-one-out method. The compounds HMF, methylnicotinate, 2-furylmethanol, acrylamide, chlorogenic acid, and lactic acid were selected as discriminating variables. The error of prediction RMSEP was 0.43 °C. The scores plot showed a partitioning of the samples according to their final temperature. The development time showed no influence on the chemical composition and no differentiation between the samples could be detected [21].

## 4. Materials and Methods

### 4.1. Samples

The calibration of the PLS model was based on 40 commercially available coffee samples. The samples were described in more detail in our previous study about LC–MS/MS reference analysis [8]. For validation purposes as the independent test set, 50 coffee samples were prepared. A total of 26 samples had a roasting degree similar to commercial samples, from which 9 were roasted in a drum roaster and 33 were roasted in an IR roaster. A total of 24 samples had more extreme roasting degrees from very light to slightly burnt (black, Neapolitan-type), which were also roasted in the IR roaster.

### 4.2. Materials

Chloroform (D1, 99.8 atom% D, stabilized with silver) and tetramethylsilane (≥99.9%, for NMR spectroscopy): Carl Roth GmbH+Co. KG, Karlsruhe, Germany.

Furan (≥99%), 2-furoic acid (98%), L-asparagine, and methylnicotinate (99%): Sigma-Aldrich Chemie GmbH, Taufkirchen, Germany.

Single use filter, Chromafil Xtra PET-45/25: Macherey-Nagel GmbH & Co. KG, Düren, Germany.

NMR tubes, Deu Quant: DEUTERO GmbH, Kastellaun, Germany.

NMR tube lids, red: SP Scienceware, Vineland, NJ, USA.

Infrared roaster, Tyboon 3000: Kammerer GmbH, Remchingen, Germany.

Drum roaster, Solar Shop Roaster: Coffee-Tech Engineering, Moshav Mazliach, Israel.

NMR equipment from Bruker Biospin, Rheinstetten, Germany including:

- NMR instrument: Ultrashield 400;
- Console: Avance III-400;
- Sample head: 5 mm PASEI 1H/D 13C;
- Sampler: SampleXpress;
- Sample head cooling: BCU05;
- TopSpin, Version 4.0.9.

Mahlkönig multipurpose mill, EK 43/1: Hemro International AG, Bachenbülach, Switzerland.

MATLAB, Version 2020: The MathWorks Incorporated, Natick, MA, USA.

PLS_Toolbox: Eigenvector Research Incorporated, Manson, WA, USA.

### 4.3. Analytical Methodology

#### 4.3.1. LC–MS/MS, Method 1, Calibration Set

The analysis of acrylamide was conducted according to the standard method EN 16618:2015 using liquid chromatography in combination with tandem mass spectrometry (LC–MS/MS). For details see Lachenmeier et al. [8].

#### 4.3.2. LC–MS/MS, Method 2, Test Set

A stable isotope dilution method (SIDA) adapted from Rünz et al. [22] was performed to determine the acrylamide content of the samples. A total of 2 g of sample material (ground roasted coffee) was weighted into a 50 mL Falcon, filled up to 30 mL with double

distilled water, and 1 μg internal standard ($D_3$ acrylamide absolute) was added. The mixture was shaken for 30 sec (vortex) and then extracted for one hour at room temperature while stirring with a magnetic stirrer. After 10 sec of shaking (vortex), the mixture was centrifuged (3000 rpm for 30 min, at room temperature). An aliquot of 10 mL was taken from the supernatant and transferred to a 10 mL Falcon.

An SPE column (Isolute ENV+; 500 mg; 6 mL; Biotage Sweden AB, Uppsala, Sweden) was conditioned with 4 mL of methanol and short vacuum and equilibrated twice with 4 mL of double distilled water. After applying the sample to the SPE, the column was washed twice with 2 mL of double distilled water and vacuum is used for drying. The acrylamide was eluted with 2 mL methanol (60%) into a 15 mL Falcon, first at normal pressure, then vacuum was used to draw the column dry. The eluate was concentrated to 500–1000 μL using a vacuum centrifuge and then filled up to 1 mL with double distilled water. The solution was pipetted into a microreaction tube and centrifuged (13000 rpm for 20 min at room temperature). The supernatant was pipetted into an injection vessel (vial) and could be measured by LC–MS/MS in the multiple reaction mode (MRM). The LC column was conditioned with 10 mL methanol (60%) before measurement. A total of 2 μL of the final solution was injected into the HPLC system. Measurements were performed using a UHPLC system (Agilent Technologies 1290 Series; Agilent, Waldbronn, Germany) coupled to a Qtrap 5500 mass spectrometer (AB Sciex Germany GmbH, Darmstadt, Germany).

The precursor ion of acrylamide $[M + H]^+$ at $m/z = 72$ fragments in the collision cell to two product ions $[H2C=CH-C=NH]^+$ with $m/z = 54$ and $[H2C=CH-C=O]^+$ with $m/z = 55$. The product ion at $m/z = 55$ was used for quantification. The internal standard with triple deuteration had its precursor ion $[M + H]^+$ at $m/z = 75$ and product ion at $m/z = 58$ [22].

### 4.3.3. NMR Spectroscopy

A total of 200 mg of ground coffee was weighed out and 1.5 mL of $CDCl_3$+TMS solution was added. Extraction was then performed for 20 min on a shaking machine. The solutions were membrane filtered and 600 μL of the filtrate was pipetted into an NMR tube. The measurement was performed on a Bruker Ultrashield 400 NMR instrument. The processing of the spectra was performed automatically (window multiplication, Fourier transform, zero referencing to TMS signal, phase correction, and baseline correction). For details on NMR measurement of coffee samples, see Lachenmeier et al. [8].

### 4.3.4. Spiking Experiments

Between 5 and 10 mg of each substance was weighed out and dissolved in 1 mL $CDCl_3$. This yields solutions of 6.35 mg/mL L-asparagine, 6.66 mg/mL 2-furoic acid, 5 mg/mL furan, and 9.31 mg/mL methyl nicotinate. Then, 10 μL of the solution were added to the coffee extracts in an NMR tube.

### *4.4. Multivariate Data Analysis*

For the calibration of the PLS model, data from 40 commercially available coffee samples are available, from which NMR spectra and acrylamide contents (μg/kg) were already measured via the standard methods. For the bucketing of the NMR spectra, the ppm range 0–11 ppm was integrated into 0.01 ppm buckets and the chloroform signal between 7.21 and 7.29 ppm was removed. This results in 1042 buckets per spectrum. MATLAB software was used to build the model and includes a command ("plsregress") for PLS regressions. In addition, the PLS toolbox from Eigenvector Research, Inc. was used.

The bucketed NMR spectra (40 samples with 1042 buckets each) represent the x-data and the results of the LC–MS/MS measurements (40 samples with one acrylamide content (μg/kg) each) the y-data. Models were developed with six PLS components (PC) each and cross validated via venetian blinds (s = 10). Different PLS regressions were performed to see the influence of data preprocessing and variable selection.

Numerical methods, such as sRatio and VIP select variables by the amount of a specific value [23]. With sRatio, the ratio of explained variance to residual variance was calculated

for each variable. The higher the value, the more important was the variable for the model. The variables with lowest sRatio were excluded from the model. Then, a new PLS model with reduced number of variables was created and iterated until a desired criterion (minimal RMSECV) was reached [24]. The VIP method proceeds similarly and considers for the calculation not only the explained variances but also the weights of the variables for each PLS component. If the value is ≥1, the variable was considered important and was left in the model [25].

GA follows an evolutionary biology approach. Data subsets ("individuals" in "population") containing a desired number of random variables ("genes") were formed. For each subset, the RMSECV ("fitness") was calculated and subsets with an RMSECV above the median were removed ("selection"). The number of subsets ("population") had consequently decreased and was now replenished with combinations of the remaining subsets ("reproduction") by swapping variables from two subsets ("cross-over"). Further, individual variables in the subsets can be exchanged ("mutation"). This process was repeated until the subsets contain similar variables, which consequently had a high selectivity for the model [26].

In forward iPLS, similar to the GA method, subsets of certain size were formed and the RMSECV for each of them was calculated. The subset with the lowest RMSECV was then combined with one other subset and the RMSECV was determined for each combination. This was repeated until the RMSECV did not decrease despite the addition of a new subset [27,28].

The rPLS method uses the weighted loadings from the PLS model to determine the relevance of each variable. The weights are assumed to correlate with the importance of a variable. This method is easy to apply, since only the number of PLS components has to be specified (variant specified). In the suggested variant, the method independently selects the number of PLS components; in the surveyed, the method selects a component number between one and a previously named maximum number [27,29].

*4.5. Validation*

After the PLS model has been calibrated, it must be validated. Cross-validation excludes s subsets from the calibration set with n samples and uses them for validation. In the venetian blinds method, every sth sample is factored out. This method is only suitable for unsorted data sets and not for sorted data, such as from time-dependent measurements. In leave-one-out cross-validation, each object is excluded once and a calibration is performed. This method is very computationally intensive and thus only suitable for smaller datasets [30]. The computed model was applied to the excluded samples and the RMSECV was calculated. The total error of the PLS model was calculated from the mean value of the individual RMSECV [10].

## 5. Conclusions

The explanation of the PLS model is limited due to the limited information on the samples and the small number of compounds identified. For a better understanding of the PLS model, more comparison spectra of possible compounds need to be recorded. More signals can be identified by, e.g., spiking. Of particular interest is the compound behind bucket 877 (9.35 ppm), since it was selected by all variable selection methods and had a high influence on PC2. The basis of the model is a small sample number from commercial coffees. The application to extreme roasting degrees, as here mainly light roasts, is therefore problematic. The RMSEP of all samples was 334 µg/kg, and 79 µg/kg if the light-roasted samples were excluded. A prediction of the acrylamide content of unknown coffee samples of medium to very dark roast appeared possible. Nevertheless, the model could be improved by inclusion of a larger number of reference samples spanning a higher variability of different coffee types and roast degrees.

**Author Contributions:** Conceptualization, D.W.L.; methodology, V.R., C.F., K.K., S.S., A.S. and C.M.B.-U.; software, A.S.; validation, V.R.; formal analysis, V.R.; investigation, V.R., C.F. and K.K.; resources, S.G.W.; data curation, V.R.; writing—original draft preparation, V.R. and D.W.L.; writing—review and editing, A.S., K.K., C.F., S.S., C.M.B.-U., E.R. and S.G.W.; visualization, V.R.; supervision, E.R. and S.G.W.; project administration, D.W.L.; funding acquisition, E.R. and S.G.W. All authors have read and agreed to the published version of the manuscript.

**Funding:** This research received no external funding.

**Institutional Review Board Statement:** Not applicable.

**Informed Consent Statement:** Not applicable.

**Data Availability Statement:** The data presented in this study are available on request from the corresponding author.

**Acknowledgments:** The team of Coffee Consulate, Mannheim, Germany is thanked for help in conducting the drum roasting experiments.

**Conflicts of Interest:** The authors declare no conflict of interest.

**Sample Availability:** Samples are available on request from the authors.

## Appendix A

**Table A1.** Selected variables using rPLS with some assignments (m = multiplet, s = singlet, d = duplet, t = triplet, and J = coupling).

| Bucket Number | ppm | Signal | Compound |
| --- | --- | --- | --- |
| 267 | 3.16 | s | OMC |
| 393 | 4.42 | d, *J* = 12.55 Hz | OMC |
| 394 | 4.43 | | |
| 730 | 7.88 | m | |
| 752 | 8.1 | s | Furan |
| 797 | 8.55 | | |
| 842 | 9 | | |
| 844 | 9.02 | | |
| 847 | 9.05 | | |
| 859 | 9.17 | | |
| 860 | 9.18 | | |
| 873 | 9.31 | | |
| 877 | 9.35 | m | |
| 879 | 9.37 | t | |
| 894 | 9.52 | s | |
| 898 | 9.56 | | |
| 900 | 9.58 | | |
| 904 | 9.62 | s | HMF |
| 906 | 9.64 | | |
| 909 | 9.67 | m | |
| 911 | 9.69 | m | |
| 913 | 9.71 | | |
| 915 | 9.73 | | |
| 936 | 9.94 | m | |
| 943 | 10.01 | | |
| 952 | 10.1 | | |
| 966 | 10.24 | m | |
| 973 | 10.31 | | |
| 998 | 10.56 | | |
| 999 | 10.57 | | |
| 1002 | 10.6 | | |
| 1005 | 10.63 | | |
| 1018 | 10.76 | | |
| 1022 | 10.8 | | |
| 1025 | 10.83 | | |
| 1026 | 10.84 | | |
| 1029 | 10.87 | | |

**Table A2.** Results of the test set of 50 coffee samples (IR = infrared roaster, A = *Coffea arabica*, R = *C. canephora*, and blend (mix) with A, Catuaí).

| Sample Id. | Roaster Type | Roasting Degree | Coffee Type | Acrylamide (µg/kg) | | | |
|---|---|---|---|---|---|---|---|
| | | | | Method 1 | Method 2a | Method 2b | PLS Result |
| 467 | IR | light | A, Catuaí | | 350 | 340 | 316 |
| 468 | IR | mid | A, Catuaí | | 238 | 235 | 306 |
| 469 | IR | dark | A, Catuaí | | 207 | 206 | 248 |
| 470 | IR | mid | A, Catuaí | | 153 | 146 | 204 |
| 471 | IR | mid/dark | A, Catuaí | | 125 | 125 | 219 |
| 472 | IR | mid/dark | A, Catuaí | | 134 | 136 | 215 |
| 473 | IR | mid | A, Catuaí | | 150 | 152 | 186 |
| 474 | IR | light | A, Catuaí | | 968 | 991 | 330 |
| 475 | IR | light | A, Catuaí | | 660 | 655 | 234 |
| 476 | IR | mid | A, Catuaí | | 297 | 296 | 262 |
| 477 | IR | dark | A, Catuaí | | 198 | 195 | 183 |
| 478 | IR | light | A, Catuaí | | 536 | 518 | 203 |
| 479 | IR | mid | A, Catuaí | 248 | 331 | 287 | 261 |
| 480 | IR | light | A, Catuaí | 1020 | 1020 | 1079 | 325 |
| 481 | IR | light | A, Catuaí | 1430 | 1200 | 1399 | 380 |
| 482 | IR | dark | A, Catuaí | 224 | 347 | 306 | 275 |
| 483 | IR | dark | A, Catuaí | 206 | 280 | 230 | 225 |
| 484 | Drum | light | A, Catuaí | | 344 | 393 | 191 |
| 485 | Drum | mid | A, Catuaí | | 274 | 266 | 185 |
| 486 | Drum | dark | A, Catuaí | | 301 | 361 | 31 |
| 487 | Drum | mid | A, Catuaí | 227 | 294 | 296 | 242 |
| 488 | Drum | light | A, Catuaí | 471 | 593 | 601 | 317 |
| 489 | Drum | light | A, Catuaí | 588 | 757 | 859 | 307 |
| 490 | Drum | dark | A, Catuaí | 250 | 319 | 295 | 207 |
| 491 | Drum | dark | A, Catuaí | 244 | 326 | 333 | 189 |
| 492 | Drum | | A, Catuaí | | 102 | 103 | 180 |
| 493 | IR | dark | R | | 224 | 257 | 266 |
| 494 | IR | mid | R | | 325 | 290 | 287 |
| 495 | IR | light | R | | 960 | 1089 | 286 |
| 496 | IR | dark | A, Malabar | | 245 | 218 | 239 |
| 497 | IR | mid | A, Malabar | | 254 | 305 | 307 |
| 498 | IR | light | A, Malabar | | 583 | 683 | 268 |
| 499 | IR | very dark | R | | 151 | 121 | 137 |
| 500 | IR | very dark | R | | 130 | 131 | 149 |
| 501 | IR | very dark | A, Malabar | | 177 | 187 | 94 |
| 502 | IR | very dark | A, Malabar | | 121 | 131 | 154 |
| 503 | IR | dark | Mix 20% R | | 167 | 169 | 179 |
| 504 | IR | dark | Mix 40% R | | 197 | 192 | 254 |
| 505 | IR | dark | Mix 60% R | | 211 | 195 | 278 |
| 506 | IR | dark | Mix 80% R | | 264 | 282 | 290 |
| 507 | IR | dark | R | | 199 | 199 | 207 |
| 508 | IR | light | Mix 20% R | | 527 | 576 | 295 |
| 509 | IR | light | Mix 40% R | | 603 | 706 | 281 |
| 510 | IR | light | Mix 60% R | | 1051 | 910 | 320 |
| 511 | IR | light | Mix 80% R | | 1111 | 1164 | 251 |
| 512 | IR | light | R | | 792 | 719 | 336 |
| 513 | IR | 6.5min | A, Catuaí | | 15 | 17 | 309 |
| 514 | IR | 13 min | A, Catuaí | | 1099 | 1099 | 351 |
| 515 | IR | 19.5 min | A, Catuaí | | 201 | 200 | 261 |
| 516 | IR | 26 min | A, Catuaí | | 146 | 142 | 216 |

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
