# Peer review of "Indirect Nuclear Magnetic Resonance (NMR) Spectroscopic Determination of Acrylamide in Coffee Using Partial Least Squares (PLS) Regression"

_beverages, doi:10.3390/beverages7020031_

Round 1

Reviewer 1 Report

The authors constructed PLS regression model using NMR and LC/MSMS for acrylamide concentration from rosted coffees which is very interesting topic broad readers. 

I think it is well written paper. The exprimental design and methods were good to test the hypothesis. 

Please mention that the sample numbers is not enough for multivariate statistics. The more the better.  The ratio between sample and tested variable should be mentioned and discussed with citation.

Table S2. standard deviation should be mentioned here.

Author Response

The authors constructed PLS regression model using NMR and LC/MSMS for acrylamide concentration from roasted coffees which is very interesting topic broad readers. I think it is well written paper. The experimental design and methods were good to test the hypothesis. 

RESPONSE: Thank you! 

Please mention that the sample numbers is not enough for multivariate statistics. The more the better.  The ratio between sample and tested variable should be mentioned and discussed with citation.

RESPONSE: We do not believe that there is a general agreement on a minimum number of samples and variables that might be appropriate. We also think that the current number of objects is sufficient for the purpose and that we have discussed the limitations in detail. It must also be considered that the reference analysis is quite expensive so that a larger number of samples, while potentially helpful, was not possible in this case. We have added a sentence at the end of the conclusion referring to the point that the model could be improved by adding further reference samples.

Table S2. standard deviation should be mentioned here.

RESPONSE: The Root mean square error of prediction (RMSEP) is the most appropriate criterion for judgement about multivariate regression analysis. There is no standard deviation (between samples) if this is meant here.

Reviewer 2 Report

This article discusses the indirect NMR detection of acrylamide in medium and dark roast coffee using PLS. The authors benchmark their NMR measurements against reference acrylamide concentrations from LC-MS/MS data using 40 commercial samples. They then test their model against a test set of 50 unknown samples. For medium and dark roasts, they obtained a moderately high error of 79 ug/kg. for light roasts, their model predictably performed poorly, since their training set had very few light roasts.

As a proof of principle, this article has some merit. Indeed, it is a clever idea to try to indirectly extract acrylamide concentration from NMR spectra, when the acrylamide signals are below the limit of detection for most samples. They correctly recognize that their training set was not suitable for analyzing light roast test samples. Further, they refine their method in a rigorous manner and properly validate their result. However, the large error for medium and dark roasts suggests limited applicability of this method. To alleviate this problem, a much larger training set, coupled with identification of key NMR signals influencing PC2, will be essential. Such a notable goal may even require the building of an NMR database. From their discussions and conclusions, nothing is mentioned how this method is to be further developed and improved. Overall, the large error reduces usefulness and overall the impact of this paper.

I also have several specific comments:

The last sentence in the abstract is not correct. The light roasts had to be excluded solely because they were underrepresented in the training set (only 2 samples). With an appropriate training set, the indirect NMR/PLS method could be applicable. This was correctly stated in the body of the paper. Please fix the abstract.

In general, the figures need some reformatting and addition of color to help the reader.

Figure 1: Please add color to panel (b).

Figure 2: (a) Please add color and consider making the points larger. A key can be added to delineate what point is what. Also, please remove most text from panel (a), as the text makes it much too busy. A couple points of interest could be highlighted with text, but keep it to a minimum. For panel (b), please also remove the text to minimize busyness. Enlarge the points, and I suggest a color heat map for the buckets. Certain Important points (buckets) could be highlighted with text.

Figure 3: Please make changes as per my comment for Figure 2b.

Figure 4: Please use color points, not a gray scale. A print-out of the paper makes the gray-scale points a little ambiguous.

Lastly, a claim is made on page 7, “sorting by roasting method shows closer predictions for the IR roaster.” What is the RMSEP for the IR-roasted coffee? I suggest a color-coded figure in the SI, in which the points are colored by roasting method.

Author Response

This article discusses the indirect NMR detection of acrylamide in medium and dark roast coffee using PLS. The authors benchmark their NMR measurements against reference acrylamide concentrations from LC-MS/MS data using 40 commercial samples. They then test their model against a test set of 50 unknown samples. For medium and dark roasts, they obtained a moderately high error of 79 ug/kg. for light roasts, their model predictably performed poorly, since their training set had very few light roasts.

RESPONSE: Thank you for evaluating our paper.

As a proof of principle, this article has some merit. Indeed, it is a clever idea to try to indirectly extract acrylamide concentration from NMR spectra, when the acrylamide signals are below the limit of detection for most samples. They correctly recognize that their training set was not suitable for analyzing light roast test samples. Further, they refine their method in a rigorous manner and properly validate their result. However, the large error for medium and dark roasts suggests limited applicability of this method. To alleviate this problem, a much larger training set, coupled with identification of key NMR signals influencing PC2, will be essential. Such a notable goal may even require the building of an NMR database. From their discussions and conclusions, nothing is mentioned how this method is to be further developed and improved. Overall, the large error reduces usefulness and overall the impact of this paper.

RESPONSE: Thank you for pointing out these limitations and potential future step. We have included this argument at the end of section 3.2.

I also have several specific comments:

The last sentence in the abstract is not correct. The light roasts had to be excluded solely because they were underrepresented in the training set (only 2 samples). With an appropriate training set, the indirect NMR/PLS method could be applicable. This was correctly stated in the body of the paper. Please fix the abstract.

RESPONSE: Thank you for pointing this out. The last sentence of the abstract was deleted.

In general, the figures need some reformatting and addition of color to help the reader.

Figure 1: Please add color to panel (b).

Figure 2: (a) Please add color and consider making the points larger. A key can be added to delineate what point is what. Also, please remove most text from panel (a), as the text makes it much too busy. A couple points of interest could be highlighted with text, but keep it to a minimum. For panel (b), please also remove the text to minimize busyness. Enlarge the points, and I suggest a color heat map for the buckets. Certain Important points (buckets) could be highlighted with text.

Figure 3: Please make changes as per my comment for Figure 2b.

Figure 4: Please use color points, not a gray scale. A print-out of the paper makes the gray-scale points a little ambiguous.

RESPONSE: Figures 1-4 were color-coded and reformatted as requested.

Lastly, a claim is made on page 7, “sorting by roasting method shows closer predictions for the IR roaster.” What is the RMSEP for the IR-roasted coffee? I suggest a color-coded figure in the SI, in which the points are colored by roasting method.

RESPONSE: The claim about the roaster was deleted (as the number of samples was very low). Therefore, no further data is necessary.

This manuscript is a resubmission of an earlier submission. The following is a list of the peer review reports and author responses from that submission.

Round 1

Reviewer 1 Report

This manuscript describes an idea that didn’t work. On could argue that publishing the results might save someone else the time and effort of trying the same thing but I doubt that anyone else would try this and, in any case, it would have to be better written.

As acrylamide cannot be measured by 1H NMR, the idea is to associate other features in spectra with the level of acrylamide in coffee samples, but this would only work if other compounds in the spectra were in some way related to the level of acrylamide. The authors do not put forward any reasons for believing that this might be the cases, rather they want to avoid the need for LC-MS analysis and would like to use the NMR data already collected for quality control. The data is analysed using partial least squares and most of the discussion is on the data used to build the model, i.e. the 40 commercially available coffee samples with both LC-MS/MS and NMR data already available. It seems that 6 PLS components are used and the data scaled to achieve the best model for this data. This is not well-written and there are some very odd phrases such as “the influence of the posterior” which I think means that the loadings for the first PLS component increase for buckets towards the end of the spectra. There is no attempt to explain this phenomenon which occurs when the data are scaled suggesting small values are producing this effect. Is the noise increasing and, if so, why? Or is there a baseline drift? Have the authors even looked to see why this occurs?

Several methods are then used to select variables just because they are available in the PLS Toolbox and this just looks as though the authors don’t quite know what they are doing. Some variables are chosen by the multiple methods and identified by spiking but the reasoning appears to be to use the method that provides fewer variables to be identified.

The scores and loadings are then interpreted, but this is for the data used to build the model which is very likely to be overfitted. In fact, only R2 values are given whereas Q2 would at least show the fit on the data left out during model building. In section 2.3 the model is applied to independent test data. Here it would be interesting to see the scores plot for the test data, not the training data. The results of the prediction are only given in the appendix, yet this is the most interesting data. This section is difficult to follow as the text says “Including all samples…” where R2 and RMSEP values are discussed again. Does this mean a new model was built using data from all 90 samples? Or, as it should be, is the previously discussed model used to predict values for the new data without any further training? Possibly the latter given the results shown in Figure 4 although this should be made clear. Unfortunately, this figure shows that, not only can high acrylamide content not be predicted, but there is no correlation between the measured and predicted values for lower levels either. If the higher levels were removed and the graph restricted to values between 0 and 400, this would become clear. The discussion therefore makes no sense.

Reviewer 2 Report

The paper entitled “Indirect nuclear magnetic resonance (NMR) spectroscopic determination of acrylamide in coffee using partial least squares (PLS) regression” offered method for the prediction of the acrylamide content of  commercially available coffee samples. The manuscript is easy to read and presents clear objectives and once again demonstrates the versatility of the magnetic resonance. The authors are clearly indicating and discussing the limitation of their work beside the novelty and the possible future perspectives.

Reviewer 3 Report

Partial least square (PLS) regression was used to model the acrylamide content in coffee and nuclear magnetic resonance (NMR) spectroscopic data. Variable selection was used and the loadings and scores were analyzed. However, the paper may not be suitable for publication. At first, more efforts are still needed in the modeling. Although PLS is one of the best method to do the modeling, spectral preprocessing is also very important to obtain an optimal model. There have been many works reported in the modeling of near infrared spectroscopy. More things can be done following these works, particularly for the variable selection. More importantly, the model in this study cannot obtain an acceptable prediction result, as shown in Fig. 4.